# Teaching a Hands-On CTF-Based Web Application Security Course

**Bogdan Ksiezopolski** [1,*] **, Katarzyna Mazur** [2] **, Marek Miskiewicz** [2] **and Damian Rusinek** [3]

1    Polish-Japanese Academy of Information Technology in Warsaw, Koszykowa 86, 02-008 Warsaw, Poland
2    Faculty of Mathematics, Physics and Computer Science, Maria Curie-Sklodowska University in Lublin, Marii Curie-Sklodowskiej 5, 20-400 Lublin, Poland
3    CyberSkiller, Pogodna 86, 20-337 Lublin, Poland
*    Correspondence: bksiezopolski@pja.edu.pl

**Abstract:** American philosopher John Dewey, in one of his most famous theories about the hands-on approach to learning, said that practical problem-solving and theoretical teaching should go hand-in-hand. This means students must interact with their environment to adapt and learn. Today, we almost take for granted that laboratory classes are an essential part of teaching science and engineering. Specific to cybersecurity, an integral piece of any training is the opportunity to work in an interactive hands-on environment: problem-solving skills are best developed in this fashion. In this paper, we present a hands-on web application security course based on OWASP Top 10 that allows students to learn through real-life experience. The virtual laboratories provided in our course simulate common vulnerabilities and issues mapped directly from OWASP Top 10, allowing students to be well-prepared for most of the critical security risks to web applications that arise in the real world. To examine how practical knowledge affects the learning experience and to measure the effectiveness of the proposed solution, we gathered learning data (such as the number of tries and the execution time for each exercise) from our cybersecurity course applied to a group of students at our university. Then, we examined correlations between students' results and gathered statistics. In our research, we made use of a CTF-based approach, which is known as a valuable pedagogical tool for providing students with real-life problems and helping them gain more practical skills, knowledge, and expertise in the cybersecurity field.

**Keywords:** cybersecurity; web app; hands-on; virtual laboratory; OWASP; curriculum; CTF

## 1. Introduction

With the growth of digital services, the demand for cybersecurity specialists has grown considerably [1]. In its report "Cybersecurity Skills Development in the EU", the ENISA lists the key problems in cybersecurity education [2]: too few cybersecurity courses in computing curricula, poor alignment between educational offerings and the demands of the labor market, little emphasis on multidisciplinary knowledge, and the prominence of theory-based education rather than hands-on training.

A survey of cybersecurity education papers noted that cybersecurity courses predominately focus on secure programming, network security monitoring, ethical hacking, human aspects, cryptography, and authentication [3]. Only a few university courses explore web security as an independent course, in which OWASP Top 10 [4] (the most critical web application security risks list) is included. Web security has become one of the crucial problems today, and the challenge is to prepare a curriculum for teaching the mentioned security issues.

In 2020, global education encountered a major problem with the response to the COVID-19 pandemic. During the pandemic, schools and universities were closed, and education moved to online learning [5]. (The use of technology in many other fields [6]

increased as well.) In such a situation, the practical teaching aspect of cybersecurity becomes a particularly difficult challenge. The major problems faced by schools and universities include preparation of an infrastructure on which students are able to legally perform hands-on tasks.

Another key aspect of learning is getting students motivated and engaged. This is especially difficult in times of a pandemic, when students spend most of their time indoors and even have to periodically endure lockdowns.

In this article, we share a detailed description of the design, course modules, and hands-on labs for a web security course based on OWASP Top 10, a list of the most critical web application security risks. Further, we explore the challenges of implementing this curriculum via remote learning due to the constraints of the recent novel coronavirus. This paper makes the following contributions:

1. We share our experiences, lessons learned, and curriculum for a web app security course based on OWASP Top 10.
2. We discuss the challenges of remote learning and our experience overcoming these challenges in a web application security course.

## 2. Related Work

To develop and train successful cybersecurity professionals, it is important to engage students in a way that bridges the gap between academia and the real world. For this reason, several solutions have been created and are offered as hands-on cybersecurity labs. In this section, individual approaches to web security topics are briefly characterized, and, finally, the major limitations of the current state-of-the art are indicated.

Chen and Tao [7] describe a tool called Secure Web Development Teaching (SWEET) to introduce web application development security. The tool provides preconfigured virtual computers in the lab with introductory tutorials, virtualized hands-on exercises, and web application security projects. The authors describe the tools used and the resources offered to instructors, and end with an example course module on cryptography.

Ernits and Kikkas in [8] introduce an Intelligent Training Exercise Environment (i-tee), which is an open-source and fully automated platform for cyberdefense classes and competitions. The platform allows educators to simulate a realistic cyberattack in a virtual and sand-boxed environment to provides students a hands-on experience of a critical situation and includes automated grading and immediate feedback using a virtual teaching assistant.

Willems and Meinel [9] present a software solution for assessing practical exercises in an online lab based on virtual machine technology. For comparison with other tools, they considered the Tele-Lab platform on which this system was implemented.

In [10], Rahouti develops a broad range of hands-on labs. In these readily available labs, students create their own prebuilt virtual machine images, in which they implement and configure all the necessary tools, software, and security/cryptographical libraries needed to run both these labs and other cryptography experiments. ReSeLa [11] is a virtual platform based on multiple VMs. This platform is introduced to provide students with remote access in order to experiment with malware and ethical hacking in a secure environment. The project is based on the following educational framework: Conceive, Design, Implement, Operate (CDIO1). The paper outlines the background of earlier efforts on similar platforms in Sweden and Latvia. It also compares the approach with lessons learned during international projects such as PlanetLab, EmuLab, and GENI.

In [12], Fernández-Caramés and Fraga-Lamas propose a practical use case-based methodology that enables the performance of industrial cybersecurity audits quickly on exposed devices on the Internet. The teaching approach described is blended and was tested at the University of A Coruña (Spain) during the pandemic lockdown.

All the approaches present hands-on cybersecurity labs. The major limitation of the above-mentioned approaches is that the web application security domain is mentioned only in the curriculum. Another key limitation is the lack of additional mechanisms that

would serve to increase student motivation and engagement, such as CTF-based [13,14] approaches.

### 3. Course Overview and Architecture

This section describes the architecture, design, and implementation of the course prepared for students at our university. In preparing our solution, we considered the newest research of how to provide and implement highly durable and manageable web applications for our CTF exercises [15–17].

#### 3.1. Overview

During the course, we gave students access to hands-on cybersecurity laboratories that address OWASP Top10 2021 [4]. We planned the course to take 16 h, lasting 8 weeks, with 4 meetings being led by an academic teacher in synchronous mode. We divided the teaching process into micro-learning blocks (exercises) that can take between about 5 and 25 min to solve. To encourage students to participate actively in the course, we decided to introduce a CTF-based gamification system. Because we developed the course for online learning, students can additionally do the hands-on exercises any time at home. Every exercise provides introductory theoretical materials and videos showing how to solve the problem. Students can use the materials before they try solving the exercise themselves or when they encounter difficulties.

The proposed hands-on labs focus on vulnerabilities and issues from OWASP Top 10, a list of the most critical web application security risks. We mapped these 10 security risks into eight labs.

- **Lab1: Introduction to Web Application Security** (OWASP A02:2021—Cryptographic Failures, OWASP A04:2021—Insecure Design)
- **Lab2: User Authentication** (OWASP A07:2021—Identification and Authentication Failures, OWASP A03:2021—Injection, OWASP A02:2021—Cryptographic Failures)
- **Lab3: Function and Data Access Control** (OWASP A01:2021—Broken Access Control)
- **Lab4: SQL Injection** (OWASP A03:2021—Injection)
- **Lab5: Cross-Site Scripting (XSS)** (OWASP A08:2021—Software and Data Integrity Failures)
- **Lab6: Handling Data from an Untrusted Source** (OWASP A09:2021—Security Logging and Monitoring Failures, A10:2021—Server-Side Request Forgery)
- **Lab7: Processing of Composite Data** (OWASP A08:2021—Software and Data Integrity Failures)
- **Lab8: Configuration Errors** (OWASP A05:2021—Security Misconfiguration, A06:2021—Vulnerable and Outdated Components)

#### 3.2. Architecture

To implement a course such that students are able to run the exercises in their own isolated laboratories without interfering with other students, we decided to design and develop our own lab architecture, which is a cloud-based, scalable, and cost-effective solution fully accessible both at the university and at home.

A web-based portal was provided to allow students to start the exercises and submit solutions. To access hands-on lab exercises, a student should log in to a web-based portal, download his/her own VPN configuration file, and connect to a VPN server. Next, the exercise is deployed as a container to lower the cost and to speed up the starting process; it is attached to a network interface accessible only to the selected students who start the exercise. We designed exercises in a CTF (Capture the Flag) style: students solve the exercise to get the flag, which is then submitted in the web portal. After submitting the flag, users gain points; they can see their ranking on the leaderboard, compare themselves with other learners, and receive feedback in areas where they should improve. This helps them enhance motivation and set specific goals based on their level on the leaderboard.

## 4. Course Implementation

As we have shown in the related-work section, numerous ongoing studies are examining the application of gamification design methodology and game mechanics to learning environments. Most of them state that gamification in education enhances learners' motivation, problem-solving abilities, decision-making abilities, and social skills such as communication. This is why we decided to implement a course as a semester-long CTF competition.

Each of the eight topics in the course consists of a different number of exercises that are released by the instructor every week. Each exercise is assigned a different number of stars to rank its difficulty level (one star ⋆ is the easiest; five stars ⋆ ⋆ ⋆ ⋆ ⋆ is the most difficult). Depending on the difficulty of the task and the time spent on completion, students receive points. They can see their ranking on the leaderboard and compare themselves with other students in the same group or with students in all groups. Every exercise is focused on understanding the topic, and thanks to CTF-based style, additional materials encourage learners to undertake new challenges without fear of failure.

*Course Topics and Labs*

**Lab 1: Introduction to Web Application Security.** The knowledge of how web applications work is a crucial part of understanding how to secure them. Because OWASP Top10 focuses on testing the security of web applications and web services, we decided to have some introduction exercises in our course that make it easier for students to start learning about the security of web applications. In this topic, we discuss the creation of an environment for testing the security of WWW applications and for performing basic tasks such as data preview and modification of the transmitted HTTP requests. Virtual laboratories in this topic are based on OWASP A02:2021—Cryptographic Failures and OWASP A04:2021—Insecure Design and consist of seven exercises described in detail below.

*Exercise 1.1: Response Headers Preview* (⋆). This lab introduces students to the concept of HTTP headers. HTTP headers are key–value pairs that let the client (your web application) pass additional information back to the server and accept the same type of data from the server. For that reason, a good cybersecurity specialist should be familiar with essential HTTP headers. In this exercise, students preview headers of server responses using developer tools of a chosen browser to obtain the solution to the exercise. This lab is the foundation for more complex web application-based tasks. Such an exercise can be recreated using any website and any web browser that supports header previewing.

*Exercise 1.2: Manipulating HTTP Parameters* (⋆). This task is about manipulating the ID parameter of the application to find some hidden links and information on the website. After reaching the right entry, students receive the solution to the task. By solving this exercise, students can test different parameters of the web application. The knowledge of how to modify application parameters can be crucial while trying to find vectors of attack. Different open-source solutions exist that allow recreating such an exercise: among others, DVWA [18], bWAPP [19], and Juice Shop [20].

*Exercise 1.3: Launch and Configuration of Proxy in aBbrowser* (⋆⋆). In this exercise, students are familiarized with the concept of a proxy. They use and configure the two most popular proxies: Burp Proxy and OWASP ZAP. The use of an intercepting proxy for web app pentesting is incredibly important. Web applications do so much more than they appear to on the surface. A single click can generate dozens of requests in the background and provide information that we never see just by browsing a site. Any one of the requests could have a security flaw in it, but if we only use our browser, we do not know that these issues are occurring. Proxies allow us to manually inspect traffic as it leaves and returns to the browser, to look back at the history of requests, and to manage rules and filters that change requests and responses on the fly. This is why it is essential for a security analyst to know how they work. This exercise can be recreated very easily because it needs only Burp Proxy and OWASP ZAP installed on a local computer.

*Exercise 1.4: Automatic Application Scan* (⋆).The major goal of this task is to find some information in the data returned by the automated scanner. Students gain knowledge of how to use the automated scanner for penetration testing and scanning for vulnerabilities. This exercise uses only OWASP ZAP installed on a local computer; thus, it can be re-created with ease.

*Exercise 1.5: Modification of HTTP Requests* (⋆). Modifying HTTP requests allows finding vulnerabilities that can be further exploited (Figure 1). This exercise focuses on setting the User–Agent header content to some specified value to obtain the solution to the task. Students learn how modification of headers, their values, and even their order can influence the behavior of the web application and the responses it returns. To recreate this task using open-source solutions, we can use Burp Suite, OWASP ZAP, or web browser tools. A website that shows the content of HTTP headers is also useful.

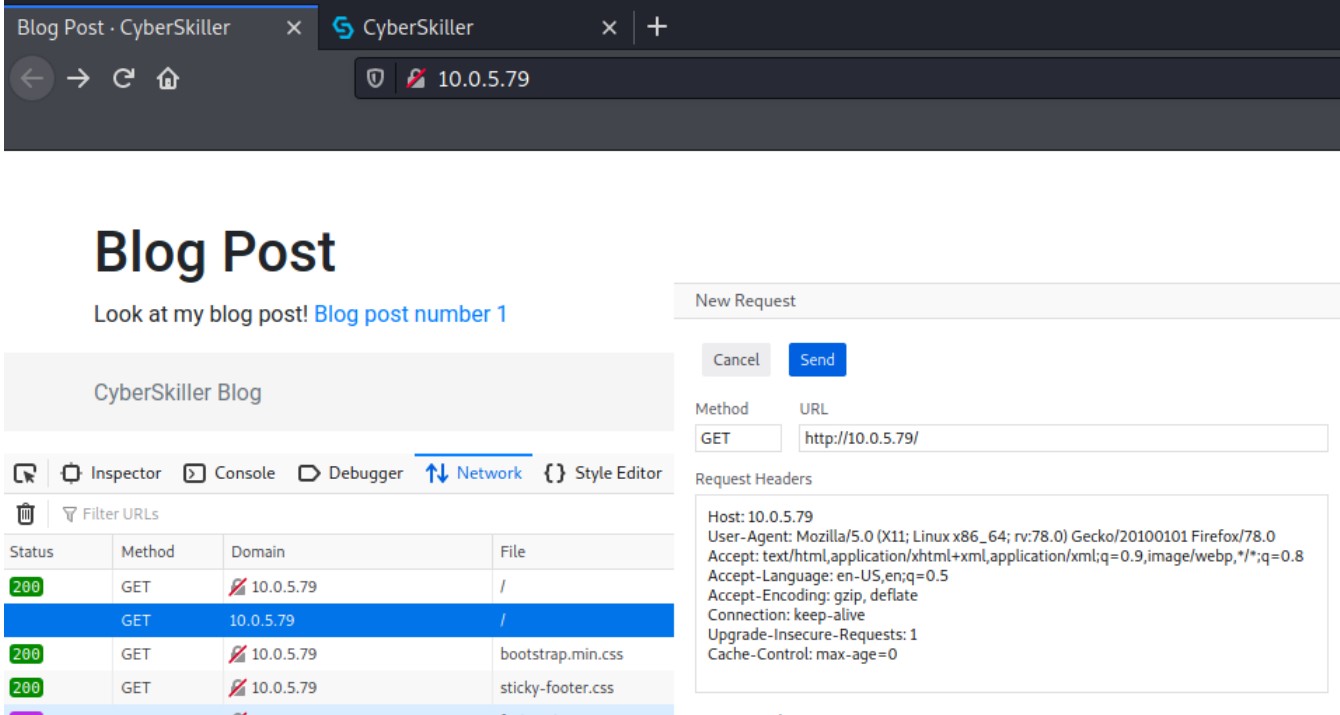

**Figure 1.** Exercise 1.5: Modification of HTTP Requests (⋆).

*Exercise 1.6: Repeating HTTP Request* (⋆). Sometimes there is a need to repeat the same request many times. It is a good idea to automate such a task. In this exercise, students use the HTTP request repeating function from Burp Suite or OWASP ZAP to resend the HTTP request to the server. We find the solution to the exercise in the server's response headers. Recreating this task using open-source tools requires the previously mentioned applications plus a website that shows the content of HTTP headers (just to check if everything worked as desired).

*Exercise 1.7: Finding the Right Parameter Value by Brute Force Method* (⋆⋆). The primary goal of this task is to learn how to use the Intruder function from Burp Suite or Fuzzer from OWASP ZAP to perform a brute force attack on the ID parameter of the application. One of the displayed entries will include the solution to the task. This lab provides an understanding of server responses and brute force attacks and can be easily recreated in the same way as in exercise 1.6.

**Lab 2: User Authentication.** This topic concerns authentication-related attacks. Authentication describes the procedure to verify one's identity. On most websites, it is encountered in the form of a username and password combination that is needed to log

in. Session management, on the other hand, comes into play when we are successfully authenticated. Upon login, a unique session key is generated. This unique key ensures that our logged-in session is held upright as we browse the application so we do not have to re-authenticate each time we switch the endpoint. Broken authentication denotes that there is an issue with the authentication or the way that the session is handled. In this module, students can detect broken authentication using manual methods and can exploit them using automated tools with password lists and dictionary attacks. They can examine and compromise session tokens. Virtual laboratories in this topic are based on OWASP A07:2021—Identification and Authentication Failures, OWASP A03:2021—Injection, and OWASP A02:2021—Cryptographic Failure and consist of five exercises, which are described in detail below.

*Exercise 2.1: Low-Complexity User Password* (⋆⋆). This lab familiarizes students with the easiest and most common way to hack into user passwords: trying the most common passwords by performing brute force and word list attacks. In the exercise, the user is given access to a web server with a simple website. To find a flag, a student must log in as an administrator. One thing a student knows about the administrator panel is that the person with administrator access does not keep a strong password. Students can use tools for automated password testing such as Hydra to test the administrator account with regard to using weak account passwords and making use of a file with a list of the most popular passwords.

*Exercise 2.2: Weak Randomness Session Identifier* (⋆⋆). After exploring password-based authentication problems, it is time to focus on session manipulation. On the server, there is a website with a login form to the administrator panel. On entering the website, an example of the correct login and password for a standard user is displayed. After logging in, the student receives a cookie operating her/his session in the portal. The name of the cookie file that operates on the session ID is given. The task is to manipulate the session cookie to gain access to the portal administrator's account. After inserting the correct authentication data, the website will display the administrative panel with the solution to the exercise. By solving this exercise, students gain knowledge of how to manage session IDs and use them to impersonate a user on the application.

*Exercise 2.3: Client-Side Authentication* (⋆). In this task, students are given access to a web server with a simple website (Figure 2). On the server, there is a website with a login form to the administration panel. The task is to successfully log in as an administrator by inspecting and/or manipulating the website's source. This exercise draws students' attention to gaps in the system, such as unsecured endpoints or client-side data validation. We can find open-source exercises similar to this one in, for example, bWAPP [19].

*Exercise 2.4: Incorrect password reset implementation* (⋆⋆). A weak password recovery mechanism is a very well known security flaw that has many task forms. This weakness may be that the security question is too easy to guess/find an answer to (question is too common, the answers can be found on media, etc.). Or there might be an implementation weakness in the password recovery mechanism code (for example, a hacker can trick a system to send a new password to an e-mail account other than that of the user). Password recovery, if not carefully implemented, can often become the system's weakest link and can be misused in a way that allows an attacker to gain unauthorized access to the system. In this exercise, students can test this vulnerability in practice. They are given access to a web server with a simple website. On the server, there is a website with a login form to the administration panel. Additionally, there are some examples of password reset links for standard users in the system. The task is to find the link to reset the administrator password, use it, and log in as the administrator to obtain the task solution. Open-source solutions that allow testing this vulnerability exist and are available in, for instance, Juice Shop [20].

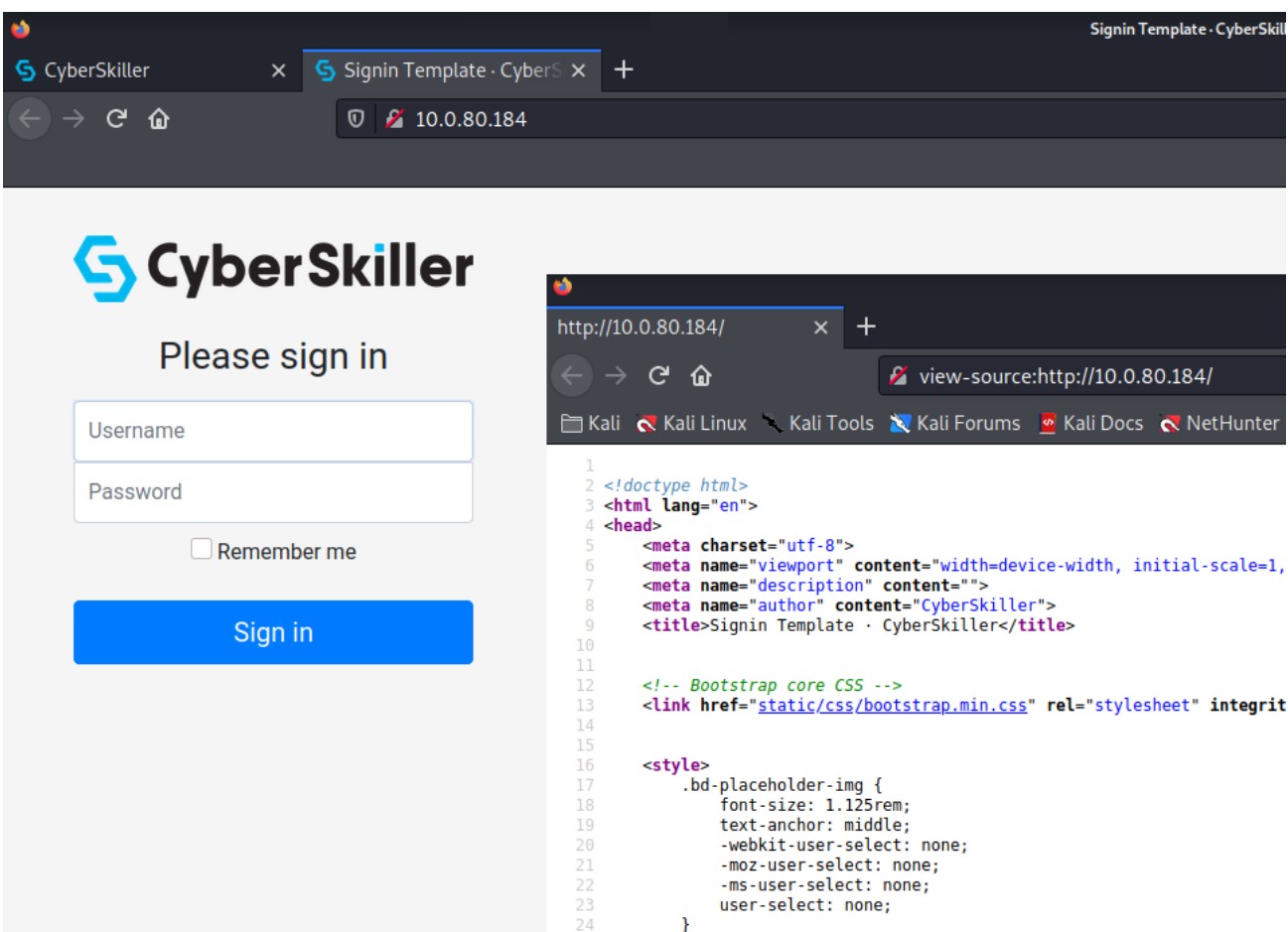

**Figure 2.** Exercise 2.3: Client-Side Authentication (⋆).

*Exercise 2.5: User Enumeration Based on Response Time* (⋆⋆). In some web applications, an attacker can retrieve potentially sensitive information by observing the normal behavior of the response times. It is possible to discover vulnerabilities in the security of a computer or network system by studying how long it takes the system to respond to different inputs. This exercise familiarizes the students with timing attacks. Students should find correct user accounts in the system by enumerating it for the occurrence of the most popular user names from the given dictionary. A flag can be found in the server's response headers. Time-based attacks can also be tested in open-source solutions such as bWAPP [19].

**Lab 3: Function and Data Access Control.** In this module, students are introduced to the weaknesses and vulnerabilities available in broken access control. Access control, also known as authorization (not to be confused with authentication), is a process that determines whether users can gain access to a resource. Authorization is a basic security service that appears in most applications. Decisions regarding access control are generally enforced on the basis of rules (called policies) set down by the user. Virtual laboratories in this topic are based on OWASP A01:2021—Broken Access Control and consist of the five exercises described in detail below.

*Exercise 3.1: Access to Hidden Pages* (⋆⋆). One of the primary steps in attacking a web application is enumerating hidden directories and files (Figure 3). Doing so can often yield valuable information that makes it easier to execute a particular attack, leaving less room for errors and wasted time. This exercise presents students the two most popular web content scanners: dirb and gobuster. The role of the student in this exercise is to search main server paths to discover an admin panel address, which contains the task solution. A similar open-source solution that deals with web content scanners can be found, for example, in the popular Juice Shop [20] application from OWASP.

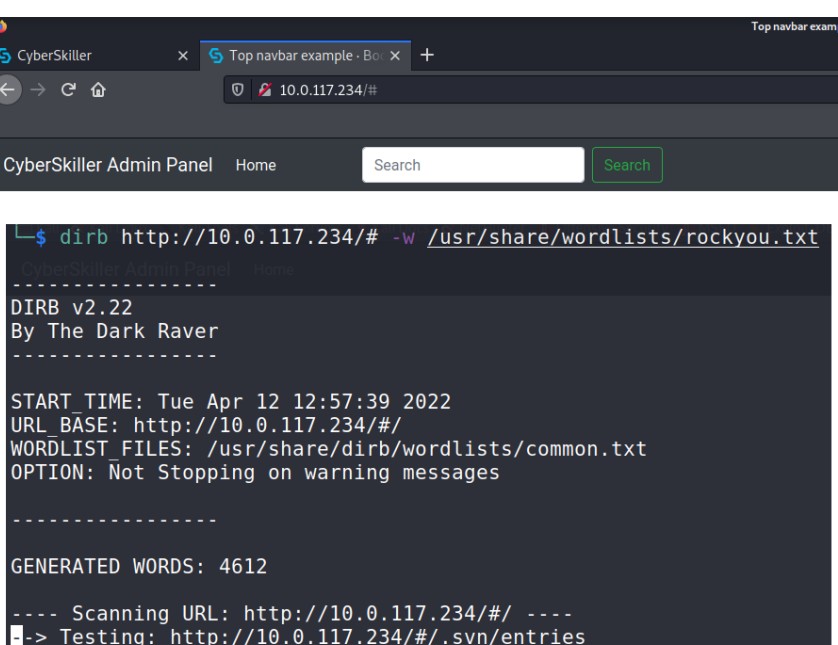

**Figure 3.** Exercise 3.1: Access to Hidden Pages (⋆⋆).

*Exercise 3.2: Security Flaw in Access to API* (⋆⋆). Proper API security ensures that user data are protected both at the application and in transit. Therefore, API security is also a critical aspect of security testing. The application in this exercise has the user search feature implemented. The website code includes a URL link for the API used to gather user data. By using this link and modifying its parameters, the student should create a request to remove the user from the system to get the task solution. The solution is given as a server response when a correct request is created. Some open-source API challenges are available in the Juice Shop web app [20].

*Exercise 3.3: HTTP Parameter Manipulation* (⋆⋆). Manipulating the data being exchanged between the browser and the web application to an attacker's advantage has long been a simple but effective way to make applications do things in a way the user often should not be able to. In a badly designed and developed web application, malicious users can modify things such as prices in web carts, session tokens, or values stored in cookies and even HTTP headers. In this task, students deal with HTTP parameter manipulation. The tested web application is an online store with basic shopping cart functionality implemented. Adequately manipulating the shopping cart's hidden parameters, such as the quantity of purchased goods and their price, leads to a situation where the cart value is negative. In such a case, the server returns the task solution in the cart. Recreating HTTP parameter manipulation with open-source applications is possible; some exercises can be found in DVWA [18] and bWAPP [19].

*Exercise 3.4: Path Traversal Vulnerability* (⋆⋆). In this lab, the application has the functionality of a thumbnail creator. Using the path traversal vulnerability of files loaded for preview leads to the loading of file *secret.txt*, situated in the parent directory of the invoked script. This file includes the flag. In this lab, students should trick the web application or web server on which the application is running. The goal is to get access to the originally inaccessible file on the web server.

*Exercise 3.5: Insecure Direct Object Reference Vulnerability* (⋆⋆). This vulnerability refers to the situation when a reference to an internal implementation object, such as a file or database key, is exposed to users without any other access control. In such cases, the attacker can manipulate those references to get access to unauthorized data. In this task, students have a chance to explore this attack in greater detail. They are given access to a WWW application that shares files with logged-in users. To get the flag, students must use the insecure direct object reference vulnerability to open secret.txt fie, which is blocked.

**Lab 4: SQL Injection.** Injection attacks are discussed in the OWASP injection module. According to the OWASP authors, injection flaws are very prevalent and are often found in SQL, LDAP, XPath, or NoSQL queries, OS commands, XML parsers, SMTP headers, expression languages, and ORM queries. They can be easily discovered using automated tools such as scanners and fuzzers. In the exercises prepared for this topic, we mostly focused on SQL-based attacks. SQL injection is an attack that inserts (injects) a malicious part of an SQL query to a database in a loaded request that is created by an application. Virtual laboratories in this topic are based on OWASP A03:2021-Injection and consist of the five exercises described in detail below.

*Exercise 4.1: Classic SQL Injection Vulnerability* (⋆). This lab embraces a classic SQL injection vulnerability. It is crucial before diving deeply into other types of SQL-based attacks. This lab provides access to a web server handling a simple HTTP website. The application has a simple user login panel. Using the SQL injection vulnerability and knowing the user login, students should log in to the panel to obtain the task solution. By solving this task, students have the opportunity to learn how to identify if an application is vulnerable to SQLi and get to know different SQL payloads. Students can recreate this exercise using open-source resources: DVWA [18], bWAPP [19], BurpSuite, and freely available SQL payload word lists.

*Exercise 4.2: Reading the Database Schema* (⋆⋆). If a web application is vulnerable to SQL injection attacks, an attacker can extract almost any data from the database (even the names and types of columns). To find a solution to this task, students should exploit a basic search engine mechanism available on a website and use UNION SQL injection to perform an attack that will return the names of tables in the database. Similar, open-source exercises can be solved using DVWA [18] or bWAPP [19].

*Exercise 4.3: Identification of the Database Server Version* (⋆⋆). A software vulnerability is a defect in software that can allow an attacker to gain control of a system. These defects can be because of the way the software is designed or because of a flaw in the way that it is coded. Databases, such as, for example, ExploitDB, exist. These databases are a very useful resource for identifying potential weaknesses in the network and for staying up-to-date on current attacks occurring in other networks. They can be a great help when it comes to finding flaws in a specific version of the given software. In this exercise, students need to use SQL injection to extract the type of the database management system (DBMS). To do this, they can exploit a basic search engine mechanism available on a website. Knowing a database type (and ideally its version), students are able to find existing vulnerabilities and ready-to-use exploits to perform an attack on a specific DBMS. Because SQL injection is still quite a popular type of attack, many open-source solutions still support it, namely: DVWA [18], bWAPP [19], and Juice Shop [20].

*Exercise 4.4: Blind SQL Injection Vulnerability* (⋆ ⋆ ⋆). In the case of blind SQL injection, we cannot see the results of the query or the errors, but we can distinguish when the query returned a true or a false response based on the different content on the page. To address this task, students are given access to a web server handling a simple HTTP website with a basic search engine mechanism. A flag is found after a user, using the blind SQL injection vulnerability, extracts from the database the password for the username provided in the exercise. Some hints, such as the name of the user table and the names of the user's table columns, are available. Blind SQL injection vulnerability can also be tested in the open-source bWAPP [19] or DVWA.

*Exercise 4.5: Time-Based SQL Injection Vulnerability* (⋆ ⋆ ⋆). Sometimes, when trying to confirm an SQLi, there is not be a noticeable change on the page that we are testing. This indicates a time-based SQL (Figure 4), which can be identified by making the database perform actions that impact the time the page needs to load. Examining this vulnerability, students deal with a web server handling a simple HTTP website with a basic user password reset mechanism. The goal is to find a flag after extracting from the database the password for the given username. Same as in the previous exercise, the name of the user table and

names of the users' table columns are provided. Students can test this type of SQL attack in open-source bWAPP [19] or DVWA.

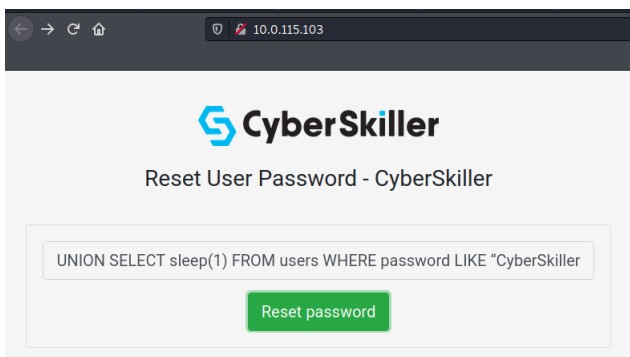

**Figure 4.** Exercise 4.5: Time-based SQL injection vulnerability lab.

**Lab 5: Cross-Site Scripting (XSS).** According to OWASP, cross-site scripting (XSS) attacks can be found in around two-thirds of all web applications. Cross-site scripting (XSS) attacks are a type of attack involving injection, where malicious input data (such JavaScripts) are inserted in the HTML code of WWW pages. There are three forms of XSS, usually targeting users' browsers: reflected XSS (injecting code to the HTTP request), stored XSS (injecting code into a data source that provides data for the page), and DOM XSS (used when an application uses JavaScript to dynamically create the page content). Virtual laboratories in this topic are based on OWASP A08:2021—Software and Data Integrity Failures and consist of the seven exercises described in detail below.

*Exercise 5.1: Stored XSS Vulnerability* (⋆). Stored XSS takes some more knowledge of the application architecture to execute successfully (Figure 5). This server-side attack originates with the website database and runs unbeknownst to the user. It is known to be the most damaging XSS code. This assignment shows students how to use the application's comments functionality (allowing commenting on an entry on a webpage) to perform a stored XSS attack.

*Exercise 5.2: Reflected XSS Vulnerability* (⋆). Reflected XSS is by far the most popular type of XSS and is the easiest to test for. In this type of XSS attack, the student does not need to know the application. All she/he must do is look for reflected values. Automated XSS vulnerability scanners can be used to find this type of issue. In this exercise, students are given access to a web server handling a simple HTTP website that uses a request GET parameter to present data to the user, at the same time placing it in the body of the HTTP response.

*Exercise 5.3: DOM XSS Vulnerability* (⋆). This XSS attack is a client-side attack only. It is a mix of persistent and reflected XSS. To gain points for this exercise, students must use the search functionality of a website (which presents to the user the search results for the entered phrase) to perform a DOM XSS attack.

*Exercise 5.4: XSS Vulnerability (Other Vector)* (⋆⋆). In this task, students are given access to a web application with comment functionality, allowing them to comment on an entry on a HTTP page. They should use it to perform an XSS attack and obtain the solution. However, the application does not allow embedding a JS script in the comment's content. The task's solution is added as a comment on a webpage if the student achieves a situation where a dialog box appears with any message. The primary goal here is to determine how comments are added on a page and to modify a POST request body to embed a JS code. In this exercise, students get familiarized with various XSS payloads. XSS attacks are implemented in open-source solutions such as bWAPP [19] and DVWA and can be used to test for various (not only JS) XSS payloads.

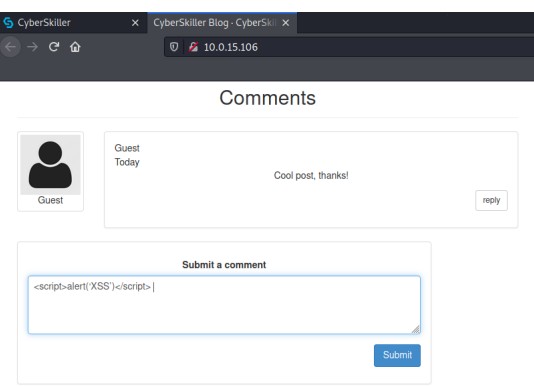

**Figure 5.** Exercise 5.1: Stored XSS Vulnerability (⋆).

*Exercise 5.5: XSS Vulnerability (Filtering Out Tags)* (⋆⋆). This exercise uses the same web application as Exercise 5.4. Here, unlike the previous one, the application has a mechanism for removing dangerous HTML tags from the entries. A solution is found if students can trick the filtering mechanism and cause the dialog box to appear with any message. Because this exercise is similar to Exercise 5.4, it can be recreated using the same open-source solutions.

*Exercise 5.6: XSS Vulnerability (Improved Tag Filtering)* (⋆⋆). For this task, the mechanism of removing dangerous HTML tags from entries is improved. Same as in the previous exercise, a solution is found if students can trick the filtering mechanism and cause a dialog box to appear with any message. Because this exercise is similar to Exercises 5.4 and 5.5, it can be recreated using the same open-source solutions.

*Exercise 5.7: XSS Vulnerability (Input Validation)* (⋆ ⋆ ⋆). This exercise is another variant of the previous exercises. However, this time, the application validates input data. Again, students find the solution if they can trick the validating mechanism and cause a dialog box to appear with any message. We can also find XSS attacks with input validation in bWAPP [19] and DVWA.

***Lab 6: Handling Data from an Untrusted Source.*** Data coming from external sources (such as data entered by application users) cannot be recognized by the application as trusted; the application should verify their correctness (e.g., format). One of the most common web application security vulnerabilities is an incorrect check of the correctness of the input data from a client or environment. Data that are modified or prepared unexpectedly can be used for application logic abuse attacks, denial of service (a DoS type of attack), or execution of any code after deserialization of the data. In this section, students learn about common security gaps that emerge from incorrect or unimplemented data validation mechanisms. Virtual laboratories in this topic are based on *OWASP A09:2021—Security Logging and Monitoring Failures* and *OWASP A10:2021—Server-Side Request Forgery* and consist of 10 exercises as described in detail below.

*Exercise 6.1: Reading an Unexpected File* (⋆). Identifying untrusted data is crucial from a web application perspective. In this exercise, students are given access to a WWW server handling a simple HTTP website. The application uses the mechanism of dynamic loading of websites passed in the GET parameter to return their content to the user. To find a solution, students should read the solution from file secret.txt placed in the exercise in the parent directory of the one containing the script for dynamic website loading. Path traversal attacks, very similar to those in this exercise, can be tested, for example, in open-source DVWA.

*Exercise 6.2: Reading an Unexpected File with the Use of PHP Filters* (⋆⋆). Local file inclusion (LFI) is one example that deals with handling data from an untrusted source. It allows an attacker to include files on a server through the web browser. This vulnerability exists when a web application includes a file without correctly sanitizing the input, allowing an attacker to manipulate the input and inject path traversal characters and include other files from the web server. In this task, the web application uses the mechanism of dynamic

loading of websites to return their content to the user. This time, however, the web application's script uses the PHP `include` command to load the file determined in the request GET parameter. The goal here is to identify and exploit LFI vulnerability.

*Exercise 6.3: Running a Malicious Command by Uploading a File* (⋆⋆). As we have already seen in a previous exercise, LFI attacks allow attackers to view the contents of several files inside a server. With LFI, it is also possible to execute shell commands directly on the remote server. In other words, LFI allows shell access. To get such access, students should exploit the application's functionality that allows uploading files defined by the user. Knowing that the solution is located in the secret.txt file in the uploads directory, they must upload the appropriate file to enable access to the secret.txt content.

*Exercise 6.4: Secure File Upload* (⋆ ⋆ ⋆). A tested web app has the functionality of uploading files defined by the user. In this exercise, students can use LFI to include a malicious file, which is then executed by the vulnerable web application. Knowing that the application is secured against uploading PHP files and that a file with a random name with the task solution is located in the uploads directory, students must upload the appropriate file that will enable reading of the task solution. To capture the flag, a good idea here is to become familiarized with web server permissions. Knowledge of HTTP requests is also useful.

*Exercise 6.5: Remote Reading of an Unexpected File* (⋆⋆). A web application considered in this task uses the mechanism of dynamic loading of websites passed in the GET parameter to return their value to the user and is protected by WAF, which filters out malicious file names. Students must exploit the application's functionality to read the task solution from a file secret.txt located in the parent directory of the one containing the script.

*Exercise 6.6: Standard Web Application Firewall (WAF)* (⋆ ⋆ ⋆). To make it more difficult for students to find a flag, a web application from Exercise 6.5 is now secured with a simple WAF that filters out malicious file names.

*Exercise 6.7: Protected Files Download (WAF)* (⋆ ⋆ ⋆⋆). This exercise is an extension of Exercise 6.6.

*Exercise 6.8: Insecure Log Browser* (⋆). Another variation of LFI attack is RCE via LFI log poisoning, which is considered in this task. Here, a web application allows browsing Apache2 users' activity server logs. The flag is located in the secret.txt file in the same directory as the Apache2 server log. Log poisoning as a common technique used to gain a reverse shell from a LFI vulnerability, and it can be used in this task to read the content of the secret.txt file.

*Exercise 6.9: Secure Log Browser* (⋆ ⋆ ⋆⋆). This exercise is a modification of Exercise 6.9. The log browser is now a secured HTTP server that expects a connection from a proxy server. Thus, this time, to find a solution, students must perform an attack using the X-forwarded-for HTTP header.

*Exercise 6.10: Sending E-mails* (⋆ ⋆ ⋆ ⋆ ⋆). Bypassing user input filtering mechanisms can sometimes be quite tricky. In this exercise, students are given access to a web application designed to send e-mail messages. This application, however, has basic user input filtering implemented. The solution to the exercise is located in a text file with a random name in the web application's root directory. The goal here is to use the e-mail sending functionality and find another vulnerability that, exploited together, allow for capturing the flag.

**Lab 7: Processing of Composite Data.** XML external entities (XXE) attacks can cause denial of service, file scans, and remote code execution that undermine the security of the system. Understanding the relationship between XML files, parsing, and weak parsing is imperative to understanding what an XXE attack is and why such an attack can put the system at risk. Virtual laboratories in this topic are based on OWASP-A08:2021—Software and Data Integrity Failures and consist of the six exercises described in detail below.

*Exercise 7.1: Unprotected Parsing of XML Files* (⋆). This exercise introduces students to a type of attack against a web application that parses XML input (XML external entity attack). To find a flag, students should create an XML input containing a reference to an external XML entity that will read a local file on a server. Doing so is possible only if we

deal with a weakly configured XML parser. Using XML's built-in external entity standard, this attack may lead to the disclosure of confidential data, denial of service, server-side request forgery, port scanning from the perspective of the machine on which the parser is located, and other system impacts. Examples of XEE attacks can be found in open-source solutions such as DVWA [18], bWAPP [19], and Juice Shop [20].

*Exercise 7.2: Denial-of-Service Attack with the Use of an XML Bomb* (⋆). In this task (Figure 6), students deal with an application that accepts data in XML format to generate greetings to users. This lab covers the concept of a denial-of-service attack that targets XML parsers. The principal goal is to use the functionalities built into the XML standard to attack the application with a Billion Laughs attack. This attack is also known as an XML bomb or, more esoterically, the exponential entity expansion attack. A Billion Laughs attack can occur even when using well-formed XML and can also pass XML schema validation. If the attack succeeds, the task response is returned, and the student receives the flag.

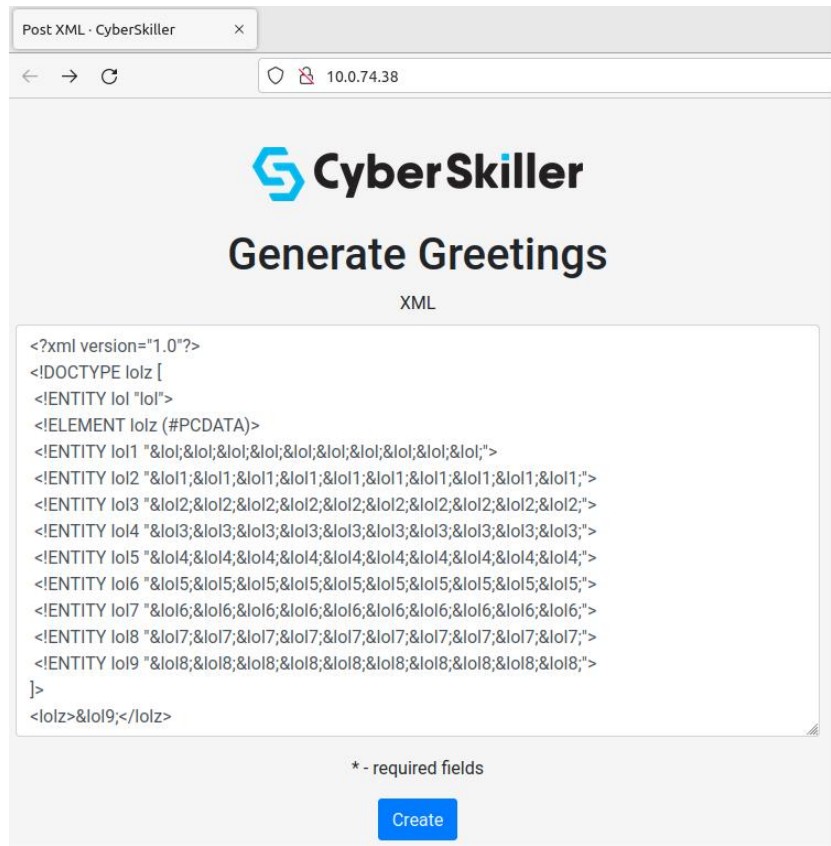

**Figure 6.** Exercise 7.2: Denial-of-Service Attack with the Use of an XML Bomb (⋆).

*Exercise 7.3: Unprotected Object Deserialization* (⋆). In this task, a web application accepts data to generate greetings to users (Figure 7). From the exercise description, we already know that data should be provided in a serialized form, which can be generated by PHP's serialization mechanism. The goal of this task is to modify the deserialized form of the greetings class to read the solution from the secret.txt file. This file is located in the parent directory of the script that handles the generation of greetings. This is how students can learn how to take advantage of deserialization weakness: when an attacker controls a serialized object that is passed into unserialize(), he can control the properties of the created object. This then allows the student to hijack the flow of the application by controlling the values passed. We can test an example of this attack using the following PHP code.

```
⋮ PHP Code Editor
 1   <?php
 2 ▾ class User{
 3       public $username;
 4       public $status;
 5   }
 6   $user = new User;
 7   $user->username = 'student';
 8   $user->status = 'not admin';
 9   $serialized_string = serialize($user);
10   $unserialized_data = unserialize($serialized_string);
11   echo $serialized_string;
12   var_dump($unserialized_data);
13   ?>
14
15   O:4:"User":2:{s:8:"username";s:6:"student";s:6:"status";s:9:"not admin";}
16   O:4:"User":2:{s:8:"username";s:6:"student";s:6:"status";s:5:"admin";}
17
```

**Figure 7.** Exercise 7.3: Unprotected Object Deserialization (⋆).

*Exercise 7.4: Protected Parsing of XML Files* (⋆⋆). Even if the application is protected against the injection of simple XML entities (XML external entity), it is still possible to perform an attack. The key to gain points for this exercise is to bypass security with external entities and read the answer to the task, which is located in the file secret.txt, located in the directory /var/www. By analyzing how to exploit lab vulnerabilities instead of just applying attack techniques using available payloads, students can have a better understating of how this attack can be performed.

*Exercise 7.5: From Deserialization of the Object to Code Execution on the Server* (⋆ ⋆ ⋆). This exercise familiarizes students with the concept of PHP POP chains. POP stands for property-oriented programming, and the name comes from the fact that the attacker can control all the properties of the deserialized object.

*Exercise 7.6: Real Attack on the Framework Using Object Deserialization* (⋆ ⋆ ⋆⋆). In this exercise, students have access to a WWW server handling a simple HTTP website. The application is based on the CodeIgniter framework, which uses object deserialization. Students should find a place where they can use deserialization and inject a chain of objects that will allow them to read the answer to the task from the secret.txt file. This file is located in the /var/www directory.

**Lab 8: Configuration Errors.** Security misconfiguration vulnerabilities can occur when a web application component is susceptible to attack due to misconfiguration or an insecure configuration option. Virtual laboratories in this topic are based on OWASP-A05:2021—Security Misconfiguration and OWASP A06:2021—Vulnerable and Outdated Components and consist of the six exercises described in detail below.

*Exercise 8.1: Publicly Accessible Administration Panel* (⋆⋆). This exercise concerns a WWW server handling a WordPress website. An admin user with a given login does not maintain the security of passwords on the website. To get access to the website, user panel students should perform a brute force attack. This exercise familiarizes students with different security tools such as wpscan, BurpSuite, and OWASP ZAP and with a collection of multiple types of lists used during security assessments. Similar open-source exercises are available in, for instance, Juice Shop [20].

*Exercise 8.2: Insecure Database Server Configuration* (⋆⋆). MongoDB is one of the most popular open-source databases. Unfortunately, this also means the ubiquity of misconfigured and unsecured MongoDB deployments. Recently, we have seen several hacks involving thousands of MongoDB databases left exposed online without protection. Usually, however, clients must connect to MongoDB servers over a network. When unsecured, this allows access by anyone. According to research, it is common for MongoDB databases to be configured to accept any connection from the Internet. The goal of this task is to connect to an insecure MongoDB server and find a solution in one of the available collections. By solving this exercise, students learn how unauthorized access works and how to check if MongoDB is exposed. Anyone can recreate this exercise by installing and configuring the MongoDB server and client on a local machine.

*Exercise 8.3: Publicly Accessible Development Server* (⋆⋆). Despite secrets such as API keys, OAuth tokens, certificates, and passwords being extremely sensitive, it is common for these to leak into git repositories through source code. Once the source code enters a git repository, from which they can organically spread to multiple locations. This includes any secrets that may be included within. Git is designed in a way that allows, even promotes, code to be freely distributed. Each time it is duplicated on git, the entire history of that project is also duplicated. This means that when you find a forgotten repository available to the public, it is easy to, for example, fetch all public commits. It is worth noticing that private repositories do not publish the source code on the internet openly, but they do not have adequate protection to store such sensitive information either. Another important consideration is that the code removed from a git repository is never actually gone. In this task, students should search a repository to find the file containing the solution. The repository is publicly accessible but hidden, so there is need to find it using some automated tools and then use already existing application that will download it to the local machine. After they have downloaded the repository, students should use built-in git tools to find the flag.

*Exercise 8.4: Using Default Passwords* (⋆⋆). As recognized by the SANS Institute, default passwords (or widely known default passwords) are one of the ten most critical Internet security threats. Default passwords are the username/password pairs that are built into the software. Software usually comes pre-installed with a standard, known username and password. This username and password are the same for all copies of a version of the software. The username is standard, and the password is initially set to the same word or character string. However, default usernames and passwords pose a serious security problem because they can be easily exploited. Exercise 8.4 examines this problem in detail: to solve this task and capture the flag, students should find the default admin password and access the application logging in as an administrator. The default password problem is also implemented in open-source solutions such as Juice Shop [20].

*Exercise 8.5: Outdated Software with Known Vulnerabilities* (⋆⋆). Using outdated software is another, still quite popular, security threat. Obsolete software creates two major issues from a security perspective:

- The similarity of exploitable vulnerabilities becomes increasingly known by attackers because the software no longer receives security updates;
- The lack of the latest security mitigation functions in older software increases the impact of vulnerabilities, making exploitation more likely to succeed (and making detection of any exploit more difficult).

*Exercise 8.6: Publicly Available Backup* (⋆⋆). This task introduces the concept of a still quite popular type of attack known as directory traversal. It aims to access files and directories that are stored outside the web root folder. By manipulating variables with reference files with "dot-dot-slash (.../)" sequences and its variations or by using absolute file paths, it may be possible to access arbitrary files and directories stored on the file system, including application source code, configuration, and critical system files. In this exercise, students already know that the administrators of the tested web application did not remove old backups from the site. They also know that the solution is located in secret.txt in one of the server's root subdirectories. To read the contents of this file, students should use a web content scanner such as dirb or dirbuster to look for backup files. Such scanners look for existing (and/or hidden) web objects and work by launching a dictionary-based attack on a web server and analyzing the response. Students capture the flag when the backup (and thus the secret.txt file) is found.

*Exercise 8.7: Browsing Public Code Repository* (⋆⋆). In this exercise, students take advantage of the fact that administrators forgot to remove the git repository during deployment. They must find secret.txt in one of the server's root subdirectories and get the contents of this file. To capture the flag, knowledge of basic built-in git tools is needed. On the other hand, by solving this exercise, students can also become familiar with different git recovery applications and scripts.

## 5. Lessons Learned

This section presents the description about data collected and shares the successes and challenges we experienced teaching a hands-on web application security course.

### 5.1. Data Collected

The data on which the statistics were based were collected from a group of 250 students in the field of computer science BSc studies at our university. The web application security course occurred during the fifth semester, which is the third year of studies (over an eight-week period from March to April 2021).

The platform on which the practical tasks during the course were performed allows data to be collected on student activity. The data were collected as records. Each record contains the following information:

- Exact time when the solving of the task started;
- Exact time when the task was completed;
- Date when the solution started;
- Date when the solution was completed;
- Time taken to solve the task (counted in seconds);
- Whether the problem was solved correctly or not (value 0 or 1, respectively);
- Name of the task;
- A number representing the difficulty level of the task;
- Name of the topic;
- Anonymized student ID.

More than 26,000 records were collected during the course mentioned, and after pre-processing, almost 22,000 records were obtained for further analysis. A deeper look at the collected data shows that the students spent almost five thousand hours working with practical tasks (4948.5 h, to be precise). The total number of attempts made was 28,296, of which 19,778 were successful. The average total time spent by a student on a task was approximately 17 h. The average number of attempts to solve a task was 71, and the average time spent per task was 14 min. Of course, the average values do not reflect the distribution of these values, as they vary over a fairly large range (Figure 8). The values close to zero and the maximum values require special attention, as they probably do not reflect what can be considered a "typical" way of solving the task.

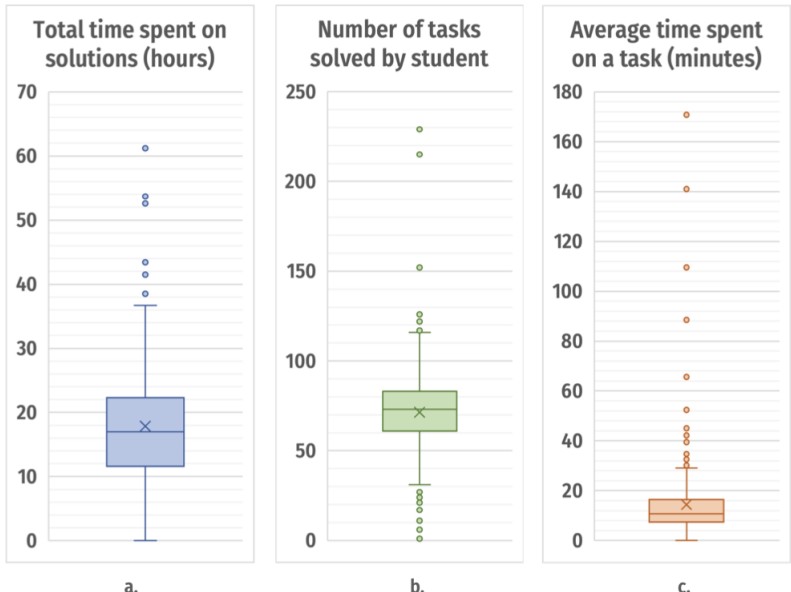

**Figure 8.** Basic statistics showing: (**a**) total time spent by the student on solving the tasks; (**b**) total number of tasks solved by the student; and (**c**) average time taken to solve the task.

*5.2. Successes*

**Students enjoy the web security course based on CTF.** At the end of the course, we asked the students to provide some feedback on the practical laboratories. The students emphasized the value of the hands-on tasks, which were much more encouraging compared to theoretical issues. When teaching online learning, and particularly the difficult topic of cybersecurity, it is important to engage students. Students stated that this form of class encouraged them to focus on cybersecurity:

*I am impressed with this way of teaching. A rather boring theory has been made great fun, which will be useful to everyone in computer science, but also in life among computers.*

*It was a great course for me. Before the first Cybersecurity class, I was very skeptical about this subject, but thanks to this platform, my perspective changed 180 degrees. It actually encouraged me to go down this career path.*

*In my opinion, this is an excellent way to carry out classes. Students can do tasks at the time they choose, and you can do them week-after-week or all at once. Everyone was really happy about this.*

**Students increase their engagement during the course.** The presented opinions are largely confirmed by analysis of student activity on the e-learning platform during the course, which was indicated by steadily increasing engagement with the assignments. This can be clearly seen in the Figure 9, which shows the total amount of time spent solving practical assignments on consecutive days of the course. This time increases significantly, with a constant number of exercises accessed weekly, which means students, in addition to solving new tasks, often gladly return to previously solved tasks.

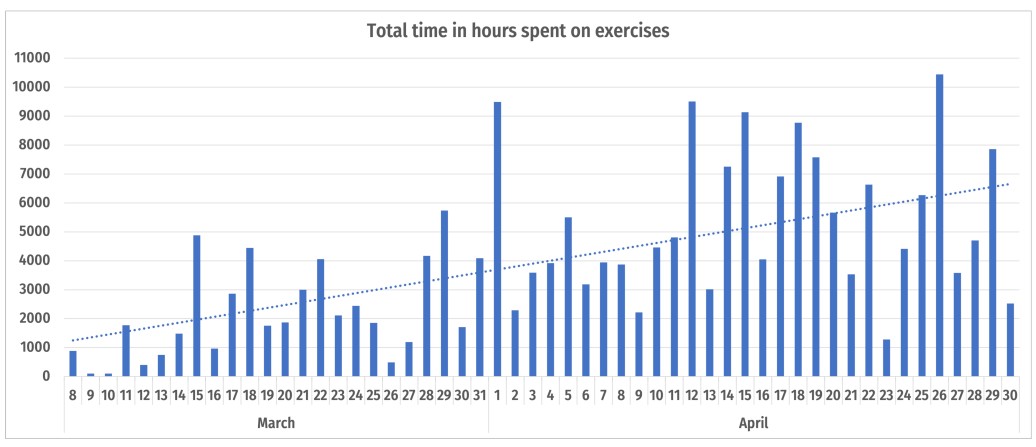

**Figure 9.** Time in hours spent by students to do the hands-on exercises.

*5.3. Challenges*

**Adjust class time based on lab difficulty.** Analysis of the time spent solving the tasks for each topic may indicate difficulties related to the complexity of the tasks (Figure 10). Tasks for the topics "Handling Data from an Untrusted Source", "SQL Injection", and "User Authentication" seem particularly time-consuming.

In contrast, tasks from topics such as "Configuration Errors", "Cross-Site Scripting (XSS)", and "Function and Data Access Control" seem to have been not so much trouble for students. This may indicate that with a fixed number of class hours per week, much more time should be allocated to difficult and time-consuming content.

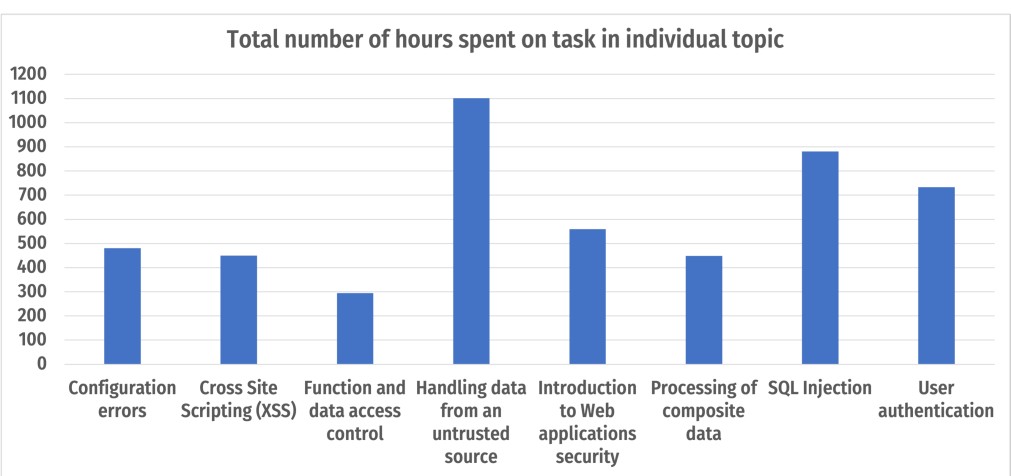

**Figure 10.** Number of hours spent on the exercises in the particular labs.

**Verification of the difficulty level of the labs.** This is confirmed by analysis of the number of attempts to solve tasks divided into successful and unsuccessful ones (Figure 11). In addition to the large number of attempts, tasks in these sections have a significant fraction of failed attempts compared to tasks in other topics.

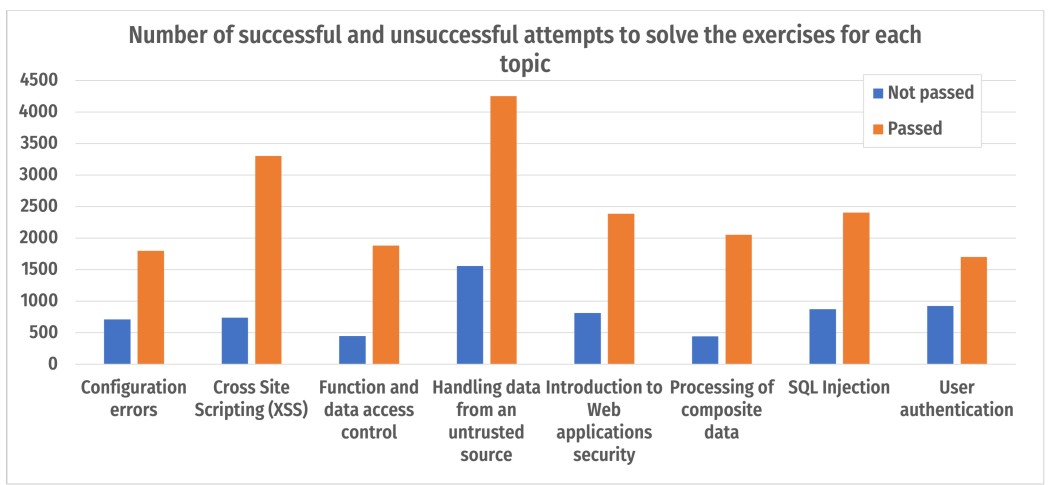

**Figure 11.** Number of successful and unsuccessful attempts to solve the exercises for each lab.

Figure 12 compares the total time spent on each task with the arbitrarily set difficulty level. For most of the tasks, the allocation seems to be correct. However, it can be seen that for some tasks, this assessment is inconsistent and needs to be verified. For example, Task 8.2 "Insecure Database Server Configuration" requires much more time to solve than the average for tasks marked with one star. This task should be marked as being significantly more difficult. Students made the greatest number of attempts at this task compared to the other tasks (760 attempts) (Figure 13), achieving the lowest success rate of about 0.5. A similar situation is observed for Tasks 4.4 "Blind SQL Injection Vulnerability" and 4.5 "Time-Based SQL Injection Vulnerability" (634 and 746 attempts, respectively). Another example can be found in Tasks 2.2 "Weak Randomness Session Identifier" and 2.4 "Incorrect Password Reset Implementation" (with success rates of 0.55 and 0.65, respectively).

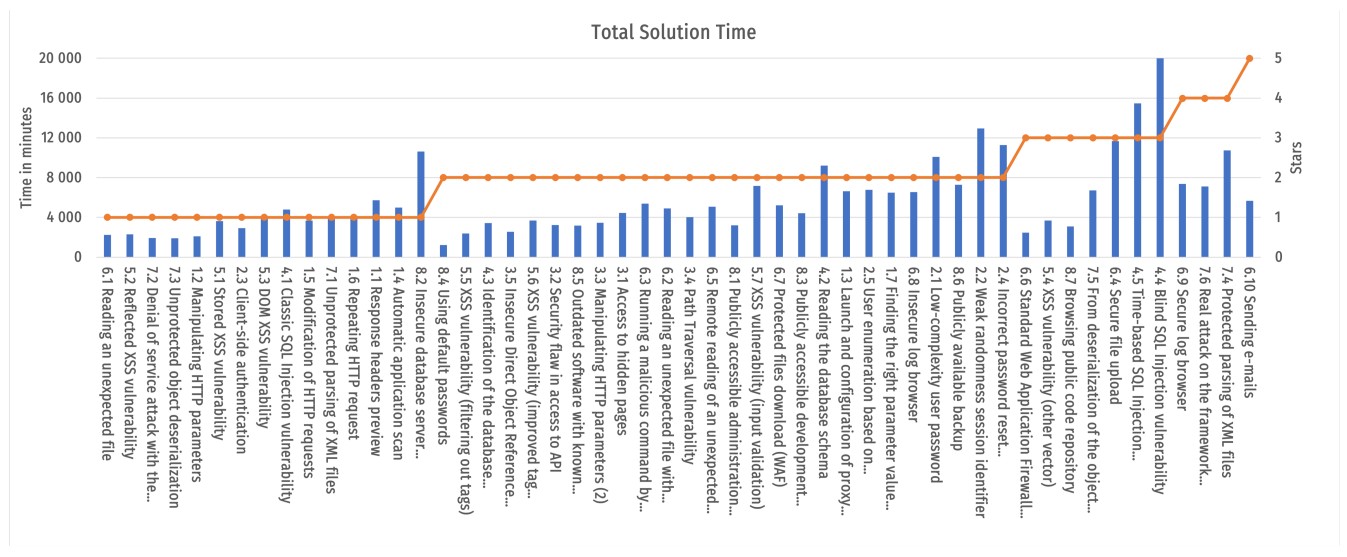

**Figure 12.** Total time spent on each task with the set difficulty level.

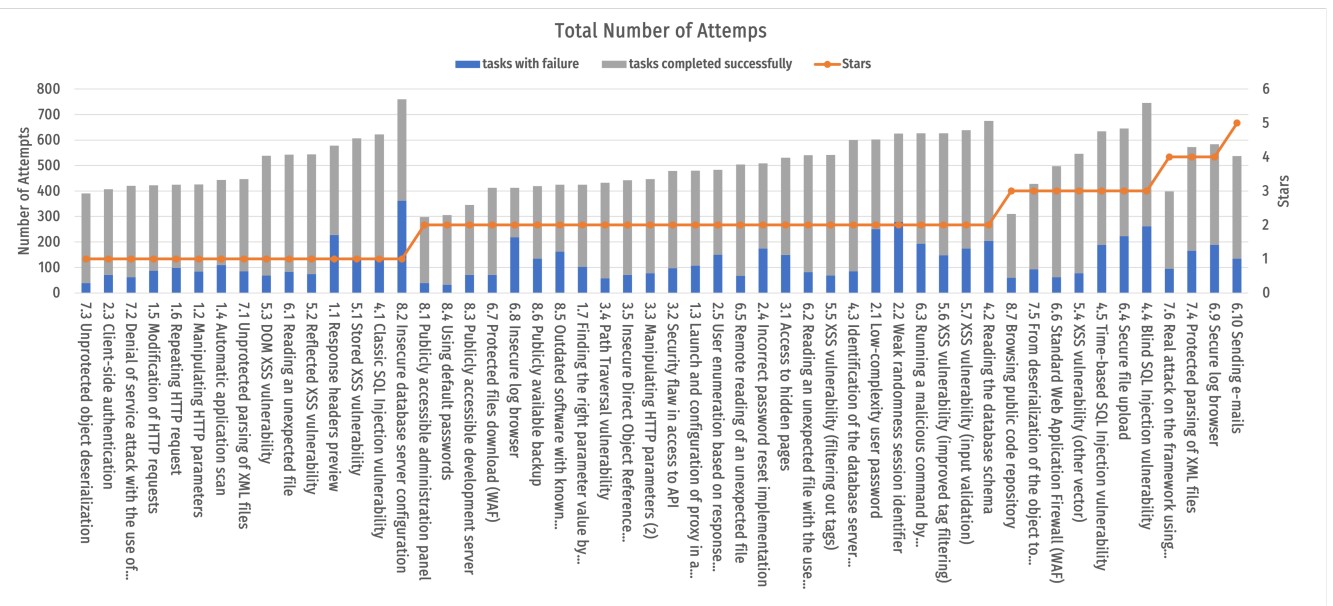

**Figure 13.** Total number of attempts taken on each task with the set difficulty level.

Analysis of Figure 14 also shows that the difficulty level of some tasks was overestimated. Task 8.4 "Using Default Passwords" seems to be easy. Students spent the least amount of time on it and achieved a high success rate of 0.9. A similar situation is in the case of Tasks 6.6 "Standard Web Application Firewall (WAF)", 4.5 "Time-Based SQL Injection Vulnerability", and 8.7 "Browsing Public Code Repository", for which the success rates were 0.88, 0.86, and 0.81, respectively. The difficulty level set for them seems to be inadequate.

Some surprises were tasks 6.9 "Secure Log Browser", 7.6 "Real Attack on the Framework Using Object Deserialization", 7.4 "Protected Parsing of XML Files", and 6.10 "Sending E-mails", which were classified as the most difficult tasks (four or five stars), while in reality they turned out to be only moderately difficult. Their average success rate was around 0.7, and they did not take much time.

In our opinion, the possibility of obtaining such detailed statistics for each task is of great didactic importance. It allows us to revise the degree of difficulty, which can be a determinant for students planning their work with the exercise material. It also influences

the student's "attitude" to the task. It helps in planning the didactic process by proper selection of subsequent tasks to constantly increase the level of difficulty.

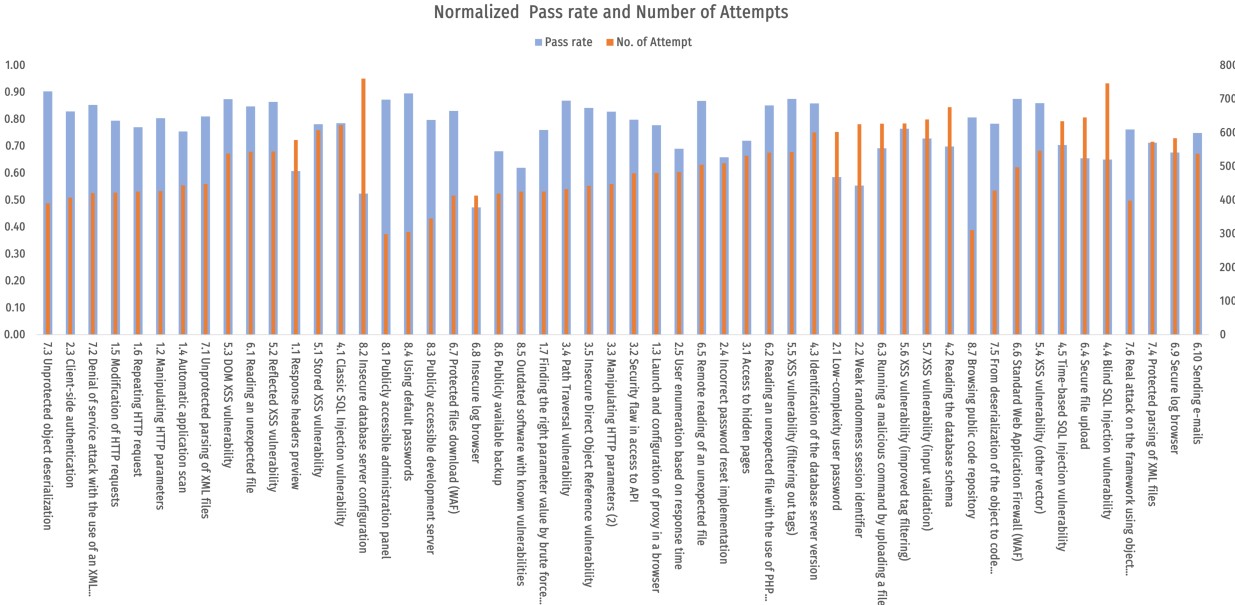

**Figure 14.** Normalized pass rate and number of attempts.

## 6. Conclusions

In this paper, we presented our web app security course based on OWASP Top 10, a list of the most critical web application security risks. We presented the full course curriculum with reference to the sites from which the virtual machines with the presented exercises can be downloaded. Our course is based on the CTF formula and thus introduces gamification to cybersecurity teaching. During the course, students devoted more and more time to problem solving. Analyzing the statistical data, it can be seen that over time, the number of correctly solved tasks significantly increased, and the average time spent on a single task decreased from week to week. These data show that the students' skills were growing over time. The article also presents students' opinions on the course. Students emphasized that practical teaching significantly increased their interest in cybersecurity. Because of the impact of the novel coronavirus, we conducted the class via a remote learning model. We have shared our experiences and hope to offer insight for instructors who wish to embrace a web security course while overcoming the challenge of remote learning.

**Author Contributions:** Conceptualization, B.K.; methodology, M.M., B.K.; software, D.R.; validation, K.M., M.M.; investigation, K.M., M.M.; resources, D.R.; writing—original draft preparation, K.M., B.K., M.M.; visualization, M.M., K.M. All authors have read and agreed to the published version of the manuscript.

**Funding:** This research received no external funding.

**Conflicts of Interest:** The authors declare no conflict of interest.

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
