# Peer review of "Teaching a Hands-On CTF-Based Web Application Security Course"

_electronics, doi:10.3390/electronics11213517_

Round 1

Reviewer 1 Report

Authors have proposed Teaching method titled as Teaching a hands-on CTF-based web application security course. Novelty of the work is low but they have presented well. Need more study and data analysis to represent the outcomes. Add more no of recent references to improve the quality of the paper and establish relevance of the topic with appropriate data.

Author Response

Dear Reviewer,

We are pleased to resubmit for publication the revised version of the article entitled Teaching a hands-on CTF-based web application security course. We appreciate your constructive criticism. We have addressed each of your concerns as outlined below, point by point.

1. Need more study and data analysis to represent the outcomes.

  • We have added Data Collected section (5.1), where we discussed these issues.

2. Add more no of recent references to improve the quality of the paper and establish relevance of the topic with appropriate data.

  • We have added 10 more references to the Introduction and Related Work sections for better relevance of the topic.

Kindly thank you for your comments in the reviews of the manuscript.

Sincerely yours,
Ksiezopolski Bogdan

Reviewer 2 Report

In this paper, the authors provide an OWASP Top-10-based hands-on web application security course that enables students to learn through practical application. The virtual labs in this course simulate typical flaws and problems that are directly mapped from the OWASP Top 10 list, ensuring that students are well-prepared for the most significant security dangers to web applications that emerge in the real world. Authors gathered learning data (such as the number of tries or execution time of each exercise) from this cybersecurity course and applied it to a group of students at a university in order to investigate how practical knowledge affects the learning experience and gauge the efficacy of the suggested solution. The outcomes of the students were then compared to the statistics that had been acquired. Authors used a CTF-based method as a useful way to teach students by giving them real-world problems to solve and helping them gain more practical skills, knowledge, and competence in the cybersecurity industry. In this form, the version is good, but the majority of the latest references are absent. Revise the whole abstract as per the objective with more intelligibility. The article's English is very poor, and several sentences are very long. There are also no comparisons between the approaches. Please update the acknowledgement and funding statements, as well. In addition, edit several references and add some key comments concerning the work's benefits and drawbacks. This manuscript can be accepted for publication after a critical revision. My other suggestion is a major and important change, kindly citing the hottest references. (2014). Risk management perspective in SDLC. International Journal of Advanced Research in Computer Science and Software Engineering, 4(3). (2020). Hesitant fuzzy sets based symmetrical model of decision-making for estimating the durability of Web application. Symmetry, 12(11), 1770. (2021). Evaluating the impact of prediction techniques: Software reliability perspective. Comput. Mater. Continua, 67(2), 1471-1488. (2022). Analyzing the Implications of Healthcare Data Breaches through Computational Technique. Intelligent Automation and Soft Computing, 1763-1779.

Author Response

Dear Reviewer,

We are pleased to resubmit for publication the revised version of the article entitled Teaching a hands-on CTF-based web application security course. We appreciate your constructive criticism. We have addressed each of your concerns as outlined below, point by point.

1. The article’s English is very poor, and several sentences are very long.

  • The paper has been proofread in terms of English. Long sentences have been shortened.

2. The majority of the latest references are absent.

  • We have added 10 more references to the Introduction and Related Work sections for better relevance of the topic.

3. There are also no comparisons between the approaches.

  • To the best of the authors’ knowledge, the literature does not describe a curriculum that would apply to the learning of web application security based on the OWSP and implemented in the CTF mode. We have write it at the end of the Related Work Section. That is why, we can not prepare the comparison, which will give the new insight in the domain.

4. Please update the acknowledgement and funding statements.

  • We have updated these information.

5. My other suggestion is a major and important change, kindly citing the hottest references.

  • We have added suggested references and other hottest references.

Kindly thank you for your comments in the reviews of the manuscript.

Sincerely yours,
Ksiezopolski Bogdan

Round 2

Reviewer 2 Report

Now, paper can be accepted.